# Review: Colon Capsule Endoscopy in Inflammatory Bowel Disease

**DOI:** 10.3390/diagnostics12010149

**Published:** 2022-01-08

**Authors:** Writaja Halder, Faidon-Marios Laskaratos, Hanan El-Mileik, Sergio Coda, Stevan Fox, Saswata Banerjee, Owen Epstein

**Affiliations:** 1Digestive Diseases Centre, Barking Havering and Redbridge University Hospitals NHS Trust, London, RM7 0AG, UK; w.halder@nhs.net (W.H.); Hanan.EL-Mileik@nhs.net (H.E.-M.); sergio.coda@nhs.net (S.C.); stevan.fox@nhs.net (S.F.); saswata.banerjee@nhs.net (S.B.); 2Wolfson Unit for Endoscopy, St Mark’s Hospital, London, HA1 3UJ, UK; 3Centre for Gastroenterology, Royal Free Hospital, London, NW3 2QY, UK; o.epstein@btinternet.com

**Keywords:** colon capsule endoscopy, inflammatory bowel disease, ulcerative colitis

## Abstract

The COVID-19 pandemic has caused considerable disruption in healthcare services and has had a substantial impact on the care of patients with chronic diseases, such as inflammatory bowel disease. Endoscopy services were significantly restricted, resulting in long waiting lists. There has been a growing interest in the use of capsule endoscopy in the diagnostic pathway and management of these patients. This review explores the published literature on the role of colon capsule endoscopy in ulcerative colitis and Crohn’s disease as a method for mucosal assessment of extent, severity, and response to treatment. Colon capsule preparation regimens and scoring systems are reported. The studies indicate that, despite inherent limitations of minimally invasive capsule endoscopy, there is increasing evidence to support the use of the second-generation colon capsule in inflammatory bowel disease evaluation, providing an additional pathway to expedite investigation of appropriate patients especially during and after the pandemic.

## 1. Introduction

The COVID-19 pandemic saw catastrophic disruption of healthcare. Most endoscopy units were restricted to investigating emergency admissions with little or no capacity for more routine investigations, such as colon screening and surveillance and the endoscopic evaluation of patients with inflammatory bowel disease (IBD). The unexpected reduction of endoscopic activity has drawn attention to the potential use of colon capsule endoscopy (CCE) to investigate and follow up patients with IBD. There is no need for day case admission, sedation, analgesia, and post procedure recovery, and the capsule can be delivered adhering to social distancing. Serious adverse events are extremely rare. While colonoscopy is the ‘gold standard’ [1], it is timely to review the potential role on the role of CCE in patients with suspected or confirmed IBD.

Small intestinal wireless capsule endoscopy was FDA approved in the United States in 2001 [2]. This orally ingested device transmits images via telemetry. CCE was first introduced in 2006 [3]. This first- generation device (CCE-1) was initially proposed as a diagnostic tool for colon cancer screening. However, a prospective study analysing the efficacy of the PillCam CCE-1 showed low sensitivity in detection of colonic polyps and a significant miss rate for colorectal cancer [4,5]. This led to the development of PillCam Colon2 (CCE-2), which provided a wider visual field using two high resolution cameras each with 172° viewing angles, improved image resolution compared to CCE-1 and an adaptive frame rate to preserve battery life, optimise image capture, and reduce reading time [6,7,8]. A European multicentre study showed that the new CCE-2 had a significantly higher sensitivity in the detection of colonic lesions [9], and was judged a suitable alternative for colorectal imaging, especially in the instances of incomplete colonoscopy [9,10,11]. The role of CCE in the investigation and management of suspected or known IBD remains to be established. CCE-2 has been used for the evaluation of both UC and CD and representative images are shown in Figure 1.

## 2. Ulcerative Colitis

In 2012, Sung et al. evaluated the efficacy of the first generation PillCam. The study recruited 100 patients with either known or suspected UC [12]. CCE-1 was compared to traditional colonoscopy in terms of its ability to evaluate mucosal healing. CCE-1 had a sensitivity of 87% and a specificity of 75%. It showed a positive predictive value of 93%, but a negative predictive value of only 65%. The authors concluded that high sensitivity and positive predictive value demonstrated the potential of CCE in the initial diagnosis of ulcerative colitis, but the 65% negative predictive value limited the evaluation of active disease. The authors reported that inadequate bowel preparation, luminal bubbles obscuring mucosal views, and in some patients, rapid colonic transit limited accurate reporting. Only 7% had excellent bowel preparation, 57% were good, 31% were fair, and 7% were poor, and it appeared reasonable to speculate that outcomes would improve with modification of bowel preparation. No capsule related adverse events were reported.

In 2013, Meister et al. reported another study using the first generation PillCam compared to traditional colonoscopy. CCE-1 assessed the mucosal disease activity and disease localisation in 13 patients with known UC [13]. The authors reported adequate bowel preparation in 90% of their patients, with a complete CCE visualisation of the mucosal surface in 80% of patients. They observed that, compared to colonoscopy, CCE offered inferior resolution of vessels, mucosal granularity, deeper damage, and extent of disease tended to be underestimated.

In 2012, Ye et al. reported a prospective study of 26 UC patients. The authors set out to evaluate CCE-1’s performance in assessing disease severity and extent [14]. CCE-1 was compared with colonoscopy, with the procedure occurring as soon as the capsule was excreted, and mucosal lesions were classified using the Baron Scale. High consistency was observed between CCE-1 and colonoscopy in the scoring of severity (k = 0.751, *p* < 0.001) and disease extent (k = 0.522, *p* < 0.001). The authors concluded that CCE-1 was able to detect delineated mucosal abnormalities in the small intestine as well as the colon, and that the image magnification of CCE-1 improved sensitivity for detecting subtle erosions and mucosal erythema. The study concluded that CCE-1 was suitable for assessing both the severity and extent of active UC inflammation.

In 2013 Hosoe et al. reported their experience of the second-generation capsule endoscopy (CCE-2) [15]. This study included 40 patients who had either mild ulcerative colitis or were in remission with 5-aminosalicyclic acid-based maintenance therapy. The images from CCE-2 were compared to the standard of colonoscopy. Expert endoscopists were blinded to the CCE results, and images were recorded using Matts endoscopic scores [16] derived from each segment of the colon, and these scores were correlated with images obtained from colonoscopy. There were strong correlation values, especially in the caecum, ascending, transverse, and left sided proximal colon, with r values ranging between 0.765–0.906, and moderate correlation in the distal part of the left sided colon (r = 0.673). The authors concluded that CCE-2 was a feasible modality for the assessment of mucosal inflammation and disease severity of UC, and therefore has a place in assessing the efficacy of treatment.

A 2014 prospective study used both the first- and second-generation colon capsule in 42 UC patients to evaluate disease activity and extent of inflammation [17], and the results compared to standard colonoscopy. CCE-2 was made available during the period when the study was conducted, thus both devices, CCE-1 (*n* = 23) and CCE-2 (*n* = 19) were evaluated. The study showed good correlations between CCE and colonoscopy in disease activity assessment (κ = 0.79; 95% confidence interval: 0.62–0.96) and extent of mucosal inflammation (κ = 0.71; 95% confidence interval: 0.52–0.90). Interestingly, in three of the 42 enrolled patients, CCE changed the diagnosis of UC to Crohn’s disease due to inflammation observed in the small bowel at CCE. This highlighted the importance of CCE in patients labelled as having “indeterminate colitis”. A limitation of this study was that most patients were in remission, undergoing surveillance colonoscopies for colorectal cancer or dysplasia, and this population is unrepresentative of the usual UC case-mix undergoing colonoscopy. In addition, CCE in this group would not normally be offered, as biopsies were required. The authors concluded that capsule endoscopy could provide an accurate estimation of UC disease severity and extent comparable to colonoscopy, and that images of the small intestine could alter the diagnosis.

In 2017, a prospective study enrolled 150 UC patients and compared CCE-2 with colonoscopy [18]. The study used both the widely used Mayo Endoscopic Scoring (MES) and the validated Ulcerative Colitis Endoscopic Index of Severity (UCEIS) for assessment of disease activity. Two experienced CCE-2 reviewers and 2 colonoscopists were blinded and, when appropriate, achieved consensus on the imaging. There was substantial agreement between CCE-2 and colonoscopy in the measurement of inflammation using either MES (intraclass correlation coefficient [ICC] 0.69; 95% CI, 0.46–0.81; *p* < 0.001) or UCEIS (ICC 0.64; 95% CI, 0.38–0.78; *p* < 0.001). CCE-2 had a high sensitivity (97%) in detecting mucosal inflammation (MES > 0), and a sensitivity of 94% and NPV of 96% in identifying moderate to severe inflammation (MES > 1). Among the three descriptors of UCEIS (vascular pattern loss, bleeding, erosions/ulcers), the sensitivity of CCE-2 in identifying vascular pattern loss was the highest. The high NPV of CCE-2 in identifying vascular pattern loss, bleeding, or erosions/ulcers in colon segments suggested that CCE-2 was reliable for ruling out inflammation. The sensitivity and specificity of CCE-2 in detecting post-inflammatory polyps was 100% and 91%, respectively. The study satisfaction survey highlighted that the majority of patients (68%) preferred capsule endoscopy to colonoscopy for monitoring their disease, which could impact on long-term compliance. The survey revealed that the cost, duration of the procedure, and the bowel preparation were the main challenges. In total, 40% of patients expressed a dislike for the taste and volume of the bowel preparation, as well as the strict diet control prior to and on the day of the procedure. The study concluded that CCE-2 accurately assessed disease severity in UC patients, was a reliable and safe tool for disease monitoring, and was the preferred investigation modality in patients [18].

In 2018, Takano et al. conducted a prospective study with 50 UC patients in clinical remission, defined by a clinical activity index (CAI) ≤ 4 [19]. CCE-2 was used to evaluate mucosal healing and disease activity using the Mayo Endoscopic Subscore (MES) and the Ulcerative Colitis Endoscopic Index of Severity (UCEIS) scores. These scores were correlated with clinical outcomes. A modified low volume bowel preparation was used, achieving total colon visualisation in 93.3% of cases, with 90% of patients excreting the capsule within 8 h. Bowel cleanliness was considered ‘good’ or ‘excellent’ in 73.3%. The rate of mucosal healing (assessed by CCE-2) was 77%, and the relapse-free rate was higher in MES 0 and 1 compared to scores of 2 and 3 (*p* = 0.04). The authors concluded that CCE-2 was acceptable for assessing the severity of mucosal disease, and that evaluating mucosal healing using CCE-2 helped outcome. A total of 70% of patients stated they would choose CCE-2 over colonoscopy for further investigations. While there was no direct comparison with standard colonoscopy, the study indicated that CCE endoscopic scores could predict clinical outcome.

In 2018, Okabayashi et al. used a simplified approach to bowel preparation in 33 patients with known UC [20]. The lower volume PEG based preparation was used, and castor oil was added to the second booster following capsule administration. There was no direct comparison with colonoscopy. Capsule excretion rate was 94% with a median colon transit time of 119 min. The MES and UCEIS scores were used to grade the mucosal inflammation, and faecal calprotectin (FC) was used to compare with the CCE-2. Interobserver agreement was found to be good with both MES (k = 0.746) UCEIS (k = 0.684) scoring systems. FC revealed a positive correlation with both MES and UCEIS scores (r values of 0.7456 and 0.7235, respectively). FC predicted a MES value of 0 (area under the curve = 0.9786). When the cut-off level of FC was set to 64 μg/g, FC below 64 μg/g predicted an MES of 0 with a sensitivity of 100%, specificity of 84.6%, positive predictive value of 90.0%, and negative predictive value of 100%. Questionnaires were used to assess patient experience with CCE-2. CCE-2 was preferred by 42.4% of patients, compared to 27.3% for colonoscopy, while 30.3% expressed no preference. The authors concluded that the castor oil regimen was well tolerated with a high excretion rate, and that good imaging was possible with CCE-2.

## 3. Crohn’s Disease

CD lesions are discontinuous and mostly involve the terminal ileum. As CCE captures both small and large bowel changes, this device might play a useful role in evaluating disease severity, extent, and distribution in Crohn’s patients [21].

In 2015, Carvalho et al. conducted a retrospective study of mucosal healing in the small bowel and colon in Crohn’s patients [22]. This small study included 12 patients with non-stricturing and non-penetrating small bowel and colonic Crohn’s disease in corticosteroid free remission with a Harvey Bradshaw Index < 5. At diagnosis, patients had undergone ileo-colonoscopy to identify active CD lesions, and small bowel capsule endoscopy to assess the Lewis Score (LS). After ≥1 year of follow-up, patients underwent entire gastrointestinal tract evaluation with CCE-2. The primary endpoint was assessment of CD mucosal healing, defined as no active colonic CD lesions and LS < 135. The majority of patients (83.3%) had received immunosuppressive therapy. In the study, the colon was segmented into caecum, ascending, transverse and descending/sigmoid colon, and rectum. Overall, two CCE pan endoscopy procedures were incomplete, the capsule reaching only the splenic flexure. In total, six patients (50%) were shown to have colonic lesions. Of these, two had ulcers throughout the whole large bowel, and the remaining four had segmental inflammatory disease. The study showed that only three patients in sustained corticosteroid-free clinical remission achieved mucosal healing in both the small bowel and the colon, highlighting the limitations of clinical disease scoring. In four patients, CCE resulted in modification of their therapy. Indices based on clinical scoring such as Harvey Bradshaw have a role to play in long term disease management, but demonstration of mucosal healing predicts fewer surgical procedures [23] and hospitalisations [24], thus mucosal imaging is currently considered a necessary investigation. The authors concluded that, as clinical remission may not reflect mucosal healing, CCE-2 may provide a safe and objective assessment of mucosal healing, helping guide the management of small and large bowel and Crohn’s disease.

In 2015, D’Haens et al. conducted a multicentre prospective study of 40 patients [25]. They all had clinically active Crohn’s disease, based on a Crohn’s Disease Activity Index (CDAI) score greater than 150, accompanied by a raised C- reactive protein and elevated faecal calprotectin. The aim of the study was to compare the endoscopic findings obtained through colonoscopy with CCE-2. Disease activity was assessed using the validated Simple Endoscopic Score for Crohn’s Disease (SES-CD) and the Crohn’s Disease Endoscopic Index of Severity (CDEIS). These patients were also required to have two or more active colonic segments. A global evaluation of lesion severity (GELS) was marked on a 10-cm visual analogue scale (VAS). Agreement between what was observed on colonoscopy and CCE-2 was analysed using the intraclass correlation coefficient (ICC). Overall, six patients had incomplete studies, due to slow transit or battery exhaustion, but the terminal ileum was observed in all the patients. Substantial agreement was observed between CCE-2 and colonoscopic assessments for CDEIS scores with an ICC of 0.65, moderate agreement (ICC = 0.50) for SES-CD, and fair agreement (ICC = 0.40) for GELS evaluations. The agreement was seen to be most significant in the ileum, with an ICC of 0.73 for CDEIS scores, with a trend of less agreement seen in the distal colon (ICC = 0.43 for left colon/sigmoid and 0.49 for rectum), which may be accounted for bowel cleanliness being worse in the distal colon. Of note, the miss rate for ulcers (false negatives) in CCE-2 was 14%. This could be explained by inadequate bowel cleansing, accelerated capsule transit, and incomplete recordings. The CCE-2 also made false positive observation in mucosal ulceration in three out of five patients, in whom there was healing at colonoscopy. This could lead to unnecessary treatment escalation in the patient. Overall, the sensitivity of CCE-2 in detecting ulcerations was 86% and a specificity of 40%, which appears to be an outsider when compared to other reports. In 15% of patients, small bowel ulceration was detected in regions of small bowel inaccessible to colonoscopy. Patient preference was assessed with 76% of patients stating that if both procedures needed repeating, then CCE-2 would be favoured. From this small study, the authors concluded that CCE-2 was safe, well tolerated, and that there was substantial agreement between CCE-2 and colonoscopy in the measurement of mucosal activity. However, further technical refinements were necessary to improve specificity.

In 2021, Yamada et al. reported the diagnostic yield of CCE-2 in 20 CD patients [26]. The study compared CCE-2 with double balloon endoscopy (DBE) as the gold standard. On the day 1, trans-oral DBE was conducted, and once strictures were excluded, the patient progressed to prepare for CCE-2. Alongside this, a patency capsule was used to exclude an obstructing stricture. On excretion of CCE-2, trans-anal DBE was performed. For analysis, the small bowel was divided in three segments (jejunum, ileum, and terminal ileum), while the large bowel was divided into four segments (right, transverse and left colon, and rectum). The study evaluated the presence of ulcer scars, erosions, and ulcers in both small and large bowel. In the study, 75% of patients excreted the CCE-2 within its battery life. Overall, five capsules were not excreted and were discovered in the large bowel. However, the terminal ileum was observed in all patients. Of 124 segments reported, the sensitivities of CCE-2 for ulcer scars, erosion, and ulcers were 83.3%, 93.8%, and 88.5%, respectively, and the specificities were 76.0%, 78.3%, and 81.6%, respectively. For the 60 small bowel segments, sensitivities were 84.2%, 95.5%, and 90.0%, respectively, and the specificities were 63.4%, 86.8%, and 87.5%, respectively. For the 64 large bowel segments, the sensitivities were 80.0%, 90.0%, and 83.3%, and specificities, 84.7%, 72.2%, and 77.6%, respectively. The relatively low specificities of CCE-2 for the detection of erosions and ulcers in the colon were thought to reflect the quality of bowel preparation, with adherent faeces often being mistaken for erosions. The authors concluded that in patients with CD, CCE-2 is well tolerated and provides a high diagnostic yield for the pan-enteric assessment of disease extent and severity. Similar to other studies, excretion rates and adequacy of bowel preparation needs to be addressed.

## 4. Performance of CCE-2 in Unselected Symptomatic Patients Subsequently Diagnosed with Various GI Pathologies, Including IBD

In 2021, Ismail et al. reported a pilot study of CCE-2 in a case-mix of patients presenting with gastrointestinal symptoms [27]. This prospective, single centre study included 66 patients who underwent both colonoscopy and CCE-2. Faecal biomarkers (faecal calprotectin and faecal immunochemical test) were also used in the analysis of these patients. This study did not focus exclusively on IBD patients, and sought instead to determine the prevalence of mucosal disease in an unselected, symptomatic cohort. The analysis recorded clinically significant colonic findings including CD, UC, and other colonic pathologies (polyps, diverticulosis, and haemorrhoids). The CCE-2 examination was complete in 76% of patients. Colonoscopy and CCE-2 revealed significant mucosal disease in 16 (24%) and 14 (21%) patients, respectively. CCE-2 detected all significant colonic polyps but detected only 43% of the patients diagnosed with UC on colonoscopy. On first read, CCE-2 revealed three patients with UC and three with CD. Missed cases were those with limited distal proctitis, due either to incomplete examination or incorrect reporting, which was recognised when the video was reassessed. During the evaluation, the booster regimen was changed to include castor-oil, and this increased excretion rates to 87% with improved bowel cleansing. Of note, seven cases were diagnosed with colitis on colonoscopy, but only three were subsequently diagnosed with histological evidence of UC. All patients with terminal ileum Crohn’s disease were detected on CCE-2. From this pilot study, the authors concluded that CCE-2 could have an important role as an alternative to colonoscopy in the investigation of intermediate-and low-risk patients with gastrointestinal symptoms, and that with improved bowel preparation and excretion rates, CCE-2 could screen for a range of mucosal diagnoses, including inflammatory bowel diseases.

## 5. Bowel Preparation

Bowel preparation in CCE is designed to optimise mucosal examination and excretion rates. For bowel cleansing, the ESGE recommended bowel preparation volume of 4 L of PEG solution, split in two doses. Small bowel boosting is necessary to optimise capsule excretion, and the ESGE regimen recommends low-dose sodium phosphate and a prokinetic in case of prolonged capsule stay in the stomach [28]. This high-volume PEG and booster regimen has proved sub-optimal results, and a common theme in the studies of CCE in IBD is the range of bowel regimens used to cleanse and boost (Table 1). This indicates the constant search for a lower volume, higher excretion regimen. Currently, the 2 L split dose PEG/ascorbate bowel cleansing regimen is often substituted for the 4 L PEG preparation and novel boosting regimens include the addition gastrografin [29], castor-oil [27] or prucalopride.

## 6. CCE Scoring Systems in IBD

In IBD, endoscopic scoring systems support objectivity and expose interobserver agreement and reproducibility of findings [31]. The scores consider many factors, including mucosal healing, colon cleanliness, and description of the lesions. With standardisation, continuity of care and decision-making regarding treatment escalation is made easier. While scores are established for colonoscopy, their application to CCE is uncertain.

For UC, the Ulcerative Colitis Endoscopic Index of Severity (UCEIS) is widely used, due its reproducibility and minimal interobserver variation [32], and the Capsule Scoring of Ulcerative Colitis (CSUC) score has been validated against UCEIS [33]. Mucosal inflammation descriptors used in UCEIS were included in the CSUC score [34]. A simple approach is applied to the calculation of the score, where the user calculates a sum of the scores of vascular patterns, bleeding, erosions, and ulcers. Matsubayashi et al. assessed CSUC in predicting future relapses in 41 patients with UC [35]. Patients in clinical remission underwent CCE-2. In the study, CSUC was higher in 12 patients who relapsed within 1 year than in 29 patients who remained in clinical remission (2.83 ± 1.95 vs. 0.72 ± 1.00, *p* < 0.01). After analysing patients who underwent CCE-2 within six months after the successful induction treatment, results showed that those with CSUC of ≤1 remained in clinical remission for a year.

Overall, two scores for evaluation of Crohn’s disease have been developed, the Lewis score and the Capsule Endoscopy Crohn’s Disease Activity Index (CECDAI or Niv score). CECDAI takes account of inflammation severity, disease extent, and presence of strictures [36]. Adapting CECDAI score to involve the colon, the novel CECDAIic allows objective assessment of the entire bowel and high agreement values were achieved amongst the five observers in the study. CECDAIic scores of less than four indicated clinical remission. The score was further studied in 2019, once again achieving excellent agreement amongst three observers, with a Kendall coefficient of 0.94 [37].

## 7. PillCam Crohn’s Capsule

Crohn’s disease may affect the entire gastrointestinal tract [38]. Therefore, the panenteric capsule endoscopy (PCE), which was developed in 2017 to assess both the small and large bowel, may provide a great deal of information regarding mucosal inflammation. The PCE is similar to the CCE-2, in that it is a two-headed capsule, giving a viewing angle of 344 °C and obtaining up to 35 frames per second with acceptable cleanliness and completeness rates in previous studies [39,40]. In 2018, a five-centre prospective feasibility study used PCE to assess 41 patients with known or suspected IBD (established CD [*n* = 29], established UC [*n* = 5] or suspected CD [*n* = 7]). The primary aim of the study was to evaluate the functionality of this new system (capsule and software) in patients with established or suspected IBD, and the secondary aims were to evaluate coverage of colon and small bowel, overall duration of reading time, and quality of images (which were graded using a Likert scale), as well as assess side effects of the procedure. As this was a feasibility study, there was no comparator against PCE. Cleansing was graded good/excellent in 95%. All 41 videos met the primary endpoint. There was no retention, and completion rate was 83%. Overall, 31% of patients with CD had proximal disease. Bowel coverage was graded 6.7 ± 0.6 and 6.1 ± 1.3 (1–7, unconfident–confident), image quality 6.1 ± 0.8 (1–7, poor–excellent), and reading time 3.7 ± 1.4 (1–7, very short to very long). There were no adverse events and notably no events of capsule retention occurred.

In 2020, a multinational, prospective study analysed 158 patients with known CD with the aim to evaluate the accuracy of PCE versus ileocolonoscopy and/or magnetic resonance enterography (MRE) for detecting active intestinal inflammation [41]. The SES-CD score was used to evaluate the TI and colon through PCE and colonoscopy, and the Lewis score was used to evaluate inflammation in the small bowel by PCE. An adapted Magnetic Resonance Index of Activity (MaRIA) scoring system was used to assess the small bowel and TI by MRE. In the small bowel, PCE had a significantly higher sensitivity (97% vs. 71%, *p* = 0.021) and specificity (87% vs. 66%, *p* = 0.020), compared with MRE. In the TI, no significant difference in sensitivity was found between PCE and the other modalities (MRE and/or colonoscopy). The specificity of CE was significantly higher than MRE combined with colonoscopy (82% vs. 37%, *p* < 0.001). In the colon, PCE and colonoscopy had no significant difference in sensitivity or specificity. There was moderate correlation in the TI (k = 0.579, 95% CI: 0.423 to 0.736, *p* < 0.001) and colon (k = 0.440, 95% CI: 0.260 to 0.619, *p* < 0.001) between PCE and colonoscopy when the SES-CD scoring index was used by both modalities. No other significant disease severity correlation was found between any of the tests. This study had only one event of capsule retention, which was managed by a second colonoscopy and stricture dilatation. It proved that PCE can be a safe, patient friendly modality in the assessment of colonic and small bowel mucosa.

Another recent multicentre study by Foong Way et al. evaluated the use of PCE for the assessment of small and large bowel mucosal inflammation in 93 patients with established (*n* = 71) or suspected (*n* = 22) CD [42]. A complete examination occurred in 85%. In this study, two cases (2.8%) of capsule retention occurred in patients with established Crohn’s disease. Disease extent was upstaged in 24 of 71 (33.8%) patients with established CD. This included patients with upper gastrointestinal or proximal small bowel disease, which was detected following panenteric examination. Disease extent was down staged in 19 of 71 (26.8%) patients, and in the remaining 29 of 71 (40.8%), disease classification remained unchanged. Interestingly, the panenteric capsule resulted in a management change in 38.7% (36/93) of patients, therefore this may be a suitable non-invasive option for determining disease activity and supporting management decisions. However, although the PCE is conceivably an appropriate endoscopic method for mapping and grading established CD, further studies are needed to support its role in a ‘treat-to-target’ strategy for disease management and monitoring [43]. In addition, preliminary research on the role of AI for the detection of erosions and ulcers using the PillCam Crohn’s capsule has shown promising results, and may not only improve accuracy, but also reduce reading times and this area merits further investigation [44].

## 8. Cost Effectiveness of Colon Capsule Endoscopy in IBD

Although several studies have investigated the cost effectiveness of CCE-2 in colorectal cancer screening [45,46,47,48], there is a paucity of publications in IBD. These have evaluated the cost effectiveness of the PillCam Crohn’s capsule in CD monitoring. Health economics is an important consideration in the management of chronic illnesses, such as IBD. A study involving 4000 simulated CD patients investigated the cost effectiveness of pan-intestinal video capsule endoscopy (PVCE) within the British National Health Service (NHS) [49]. This study estimated the annual mean cost per patient was £2191 for those receiving standard care (typically utilising colonoscopy and MR enterography for disease assessment), with a 20-year estimate of £42,266. Cost of care using PVCE for disease assessment was lower with a per annum cost of £1960, and a mean 20-year cost of £38,433. Although initial costs (in the first two years) were higher using PVCE due to the earlier initiation of biologics, in the longer term (after year three), there was a financial benefit due to reduction in surgical interventions for the management of CD.

Similar results were demonstrated in a US study, where 4000 simulated CD patients were analysed [50]. Common Monitoring Practice (CMP), which comprised of ileocolonoscopy and imaging was compared to monitoring strategies using the PillCam Crohn’s capsule. Over 20 years, use of the panintestinal capsule reduced costs ($313,367 versus $320,015), increased life expectancy (18.15 versus 17.9 years), and increased quality of life (8.7 versus 8.0 QALY [quality-adjusted life years]), making it a cost-effective option.

Despite the limitations of such simulated analyses, these studies suggested that use of the PillCam Crohn’s capsule would be cost-effective for monitoring IBD patients in the UK and USA. As yet, there is no data available on the cost effectiveness of CCE-2 in ulcerative colitis.

## 9. Conclusions

The review discussed reports on the utility of CCE-2 in the investigation of UC and Crohn’s disease. Most of the studies used colonoscopy as the “gold standard” comparator.

The studies are drawn from small or moderate sample sizes, with only a few including more than 100 patients [18]. There is little or no information on CCE reader experience, case selection varies, and there is little information on histological findings which is the true diagnostic gold standard for IBD. Bowel preparation and boosting regimens vary widely, resulting in a spectrum of excretion rates and bowel cleansing scores.

Despite the shortcomings, studies investigating UC patients has demonstrated CCE-2 to have good sensitivity and specificity for assessing extent and severity of disease, as well as interobserver agreement. Taking into account the safety and patient preference for CCE-2, available evidence suggests that capsule colonoscopy may be useful in the monitoring disease extent and severity when biopsy is not required.

Similar to UC, published reports in CD vary widely in size and design. However, there is sufficient data to indicate patient preference for CCE-2 and the potential this pan intestinal device offers when assessing extent and severity of Crohn’s disease. The passage of the capsule from stomach to rectum allows both small and large bowel imaging, and with careful pre-assessment and appropriate use of a patency capsule, capsule retention should be a rare event.

CCE-2 examination provides mucosal imaging and does not allow histological sampling. Therefore, initial diagnosis and surveillance for dysplastic changes or colitis-associated cancers is not feasible [51]. Another limitation of CCE-2 is the lack of formally trained capsule practitioners, skilled in both small and large bowel image analysis. Capsule reading times limits the number of procedures that can be assessed by a single reader, but this might soon be addressed by the use of machine learning and artificial intelligence [37]. Machine learning has been used to train computers to analyse CCE recordings in order to diagnose and assess disease severity [52,53]. In the near future, it is likely that automation will play an integral part of clinical practice in gastroenterology [54].

In conclusion, the COVID-19 pandemic has posed significant opportunities and challenges in the management of patients with chronic diseases, such as IBD. Published reports support the proposition that minimally invasive CCE-2 should have a role in the follow-up and management of patients with diagnosed IBD. It may also have important clinical utility in cases of suspected IBD where bidirectional endoscopy and small bowel imaging have not quite helped confirm the diagnosis of IBD in patients with symptoms and suggestive associations (such as iron deficiency anaemia and malabsorption). There remains a case for large scale-controlled trials to consolidate its potential role in the UC and CD pathways.

## Figures and Tables

**Figure 1 diagnostics-12-00149-f001:**
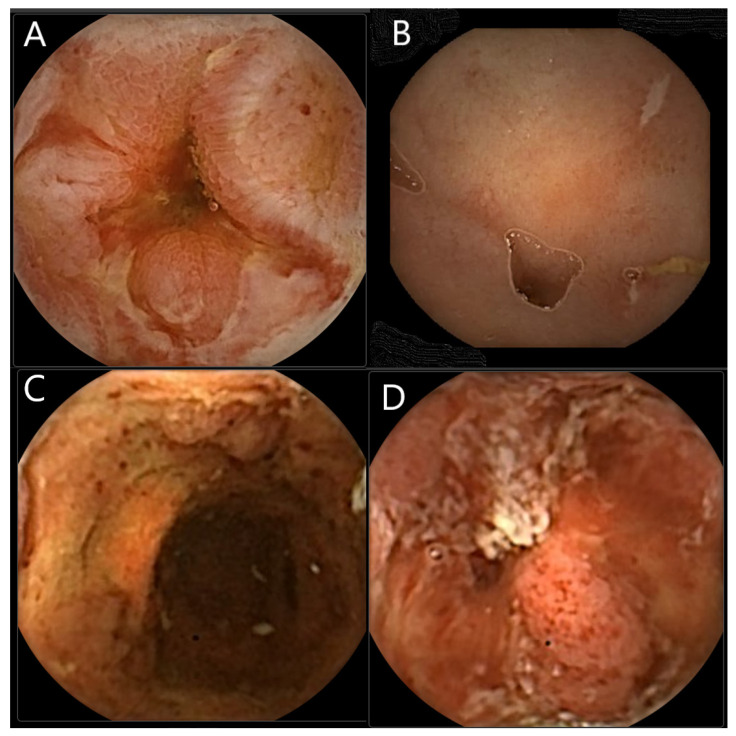
CCE-2 images of patients with IBD. (**A**) Small bowel Crohn’s disease with inflammatory structuring. (**B**). Small bowel Crohn’s disease with typical linear ulcers. (**C**,**D**). Ulcerative colitis.

**Table 1 diagnostics-12-00149-t001:** Bowel preparation regimens in studies of CCE-2 in IBD.

Study	Bowel Preparation/Prokinetic	Booster	Other Comments	Results
Hosoe et al. [10]	2 L of PEG with prokinetics—mosapride citrate and metoclopramide.	n/a	Low-residue diet the day before.	50%—excellent and good cleansing level.
Takano et al. [19]	50 g Mg citrate and 48 mg oral sennosides. 20 mg mosapride citrate.	2 × 1 L PEG/500 mL water, 50 g/180 mL magnesium citrate if capsule not excreted.	Day before—low fibre diet	73%—excellent and good cleansing level. 40% in caecum, 80% in transverse colon.
Okabayashi et al. [20]	500 mL PEG/250 mL water	Capsule in small bowel—500 mL PEG/250 mL water + 20 mL castor oil. 3 h and 6 h later—2 further boosters 500 mL PEG/250 mL water until excretion.	No restrictions the day before.	44%—excellent and good cleansing level. Acceptability (excellent to fair)—77.2%. 35.5% achieved excretion with <1l PEG.
Usui et al. [30]	700 mL PEG 20 mg mosapride citrate and 40 mg dimethicone. 10 mg metoclopramide if capsule still in stomach.	34 g Mg citrate & 20 mg mosapride citrate once capsule in small bowel. 23 g Mg citrate at 6 h if capsule not excreted.	Low fibre diet the day before.	Excellent and good cleansing level 55–65% throughout large bowel.
Ye et al. [14]	2 L PEG the evening before capsule ingestion. 1 L PEG and 50 mg itopride hydrochloride 1 h prior.	30 mL NaP once capsule passed pylorus. 15 mL NaP given at 6 h if capsule not excreted.	Low fibre diet 24 h pre procedure. Patients advised to walk after capsule ingestion	The good to excellent rate for the entire colon was 80%
Carvalho et al. [22]	1 L PEG/500 mL water the night before and morning of procedure. 10 mg domperidone if capsule in stomach.	30 mL NaP/1 L water once in small bowel. 15 mL NaP/500 mL water if capsule not excreted	Low fibre diet and 10 glasses of water 2 days before. Clear liquid diet 1 day before.	Excellent in 17% and good in 50%

## Data Availability

Not applicable.

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
