# Peer review of "Review: Colon Capsule Endoscopy in Inflammatory Bowel Disease"

_diagnostics, 2022, doi:10.3390/diagnostics12010149_

Round 1
Reviewer 1 Report
Very well written review article.
Colon capsule endoscopy is yet to take off in a big way for IBD due to all the limitations described by authors, biggest one being inability to obtain histological evidence of diagnosis, disease activity or remission, especially when colonoscopy can provide these, especially in most IBD patients other than those with involvement of small bowel proximal to terminal ileum( that cannot be reached with conventional ileocolonoscopy)
The most important utility of capsule endoscopy remains in suspected IBD in my opinion where bidirectional endoscopy +/- small bowel imaging(MRE or CTE) have not quite helped confirm the diagnosis of IBD in patients with symptoms and or suggestive associations( like iron deficiency anemia, malabsorptive picture etc)
I think if cost effectiveness comparison of colon capsule versus colonoscopy could be included in this review, it would add to the value of the paper.This is all the more important for healthcare systems like the US, where insurance companies can be difficult to get past for getting such procedures approved for these indications.
Author Response
Please see cover letter with responses to reviewers

Reviewer 2 Report
This is an important review on the role of CCE in patients with suspected or confirmed IBD. It is well-presented and presents useful information. I have minor comment. It would be better to show examples of endoscopic images by CCE and/or PCE in IBD patients.
Author Response

(The authors gave the same response as above.)
